# Non-Patent Literature

**Gema Velayos-Ortega** and **Rosana López-Carreño** *

Department of Information and Documentation, University of Murcia, 30100 Murcia, Spain;
gemavelayos@gmail.com
* Correspondence: rosanalc@um.es

**Definition:** Non-patent literature is defined as scientific publications, technical standards, conference proceedings, clinical trials, books, manuals, technical or research reports, or any other technical scientific material which is cited in patents to show what has already been published and disseminated about the invention to be patented, in order to justify its novelty. These documents are considered technically relevant to the patent granting procedure and are cited along with other patents related to the same subject matter.

**Keywords:** patents; citations; references; impact; non-patent literature; scholarly publications; NPL





## 1. Introduction

Patents, as the main exponent of technological development, contain very valuable information that is used by researchers and analysts to obtain data such as the evolution of a technology over time, the inference of institutions and companies in the development of a technological sector, the relationships between technologies, or possible future trends. However, the importance of patents is also projected in the academic–scientific field, because these documents are important indicators of scientific productivity in universities and research centres; they are mechanisms that measure the performance of scientific activity from technology transfer [1]. In the same way, the presence of scientific references in patents, and their quantification and analysis, are excellent indicators to describe this science–technology link, being key to analyse this process of technology transfer.

In this respect, numerous studies have focused on measuring the impact of these scientific references on patents, considering them to be an indicator of value. Pioneering authors in this field [2–6] have already used bibliometric procedures to quantify these data and value the transmission of knowledge from science to industry. In this way, parameters related to the degree of scientific intensity or dependence of technological sectors (by average NPL citations) were identified, as well as indicators of scientific concentration and diversification for each sector.

However, the importance of these citations also lies in aspects related to the evaluation of scientific production; according to Plaza [7], from the analysis of these citations, information is obtained about the authors/researchers, scientific institutions, and journals cited in the patents, etc. Moreover, the fact that they have been cited in these documents would add value to the influence of that publication in the technological field.

In this sense, the format and standardization of these references will become important for the establishment of metrics that allow their quantification and analysis, in order to evaluate this "technological" factor. In this way, the policies and guidelines established by each organization regarding how patent applications should be presented will be important, because it is through their procedure manuals that they establish the recommended formats for citing these documents.

On the other hand, the use of metadata and persistent identifiers associated with these references, such as DOI and ORCID, among others, will be fundamental to improve bibliographic control, because they allow the unequivocal localization of the references,

with standardised formats that can be identified by any source, in order to carry out later bibliometric analysis.

To find out more about the characteristics of these citations in the patents, it is necessary to explore the initial phase of the process of granting the patents where they are presented after the Prior Art search.

## 2. Search Reports "Prior Art": Process of Granting Patents

In the process of granting patents, the examiners responsible of their evaluation prepare a search report in which they provide previous references from scientific and technological literature, the so-called "state of the art", in order to justify the novelty and usefulness of the invention. This report distinguishes between two types of reference: on the one hand, citations to previous patents (patent literature), and on the other, references to other types of documents such as scientific articles, monographs, technical standards, among others, the so-called non-patent literature (NPL).

There are several names for this set of references, such as non-patent references (NPRs), non-patent publications and other more general terms such as non-patent citations (NPCs). Although they are all mentioned in the same way in the different research works published, the name "non-patent literature (NPL)", apart from being the most widespread, is the term most widely used by the main patent office's such as the World Intellectual Property Organization (WIPO), European Patent Office (EPO) or United States Patent and Trademark Office (USPTO).

For the preparation of this prior art search report, each organization specified a number of detailed guidelines in its manual for the patent examination procedure. In this guide, aspects of how to search for relevant documents, the sources of information to be consulted, and how to write the report are detailed. In reference to document searches, it establishes which aspects should be covered by information retrieval (justification of claims), how to formulate the search strategy, or the sources of information where the query can be made. The result of this search is set out in detail in the report accompanied by the written opinion to justify the novelty of the invention.

With regard to the authorship of the citations, they may be mentioned by the applicant/inventor himself (in the text of the patent), or by the examiner evaluating the process, which need not coincide with those provided by the applicant, who may omit them or add more references of interest to the process [8].

Furthermore, it should be noted that NPL citations do not only appear in the search reports, but are also referenced by the applicants in other parts of the patent text, on the front page, in the description of the invention, or in the claims. All these features will be evaluated to determine the relevance of these citations depending on where they have been mentioned, for what purpose, and by whom.

## 3. References Types: Taxonomy NPL

The types of documents in these references are varied and include both publications that have been evaluated and reviewed by experts and articles from scientific journals (peer review), monographs published by publishers considered to be of high quality by recognised evaluation indexes (Web of Science Book Citation Index or Scopus Book Titles), technical standards from international standardization organizations (ISO, IEC, IEEE), and other types of documents that are not so contrasted by quality standards, such as news articles, websites, or technical reports.

Depending on the technological sector to which the patent belongs, some types of documents will be used more than others, although scientific publications are usually one of the most referenced categories. This happens, above all, in the patents of technological sectors whose industry is strongly committed to scientific research and development, such as the industry derived from life sciences (biotechnology) and pharmaceutical products [9], because the productive dynamics of the research results in these scientific areas are a reference for the applied development of research.

In this line, the work of Callaert et al. [8] describes taxonomy with the different types of NPL references, in which the authors consider journal articles as a base element to establish the main categories, as can be seen in Table 1.

**Table 1.** Taxonomy. Reference types [8].

| JOURNAL REFERENCES | |
|---|---|
| SCI covered: | References of scientific publications published in journals covered in the scientific database Science Citation Index of the Web of Science, of recognized international prestige. |
| Not SCI covered: | References of scientific publications published in journals not covered in the scientific database Science Citation Index of the Web of Science |
| **NON-JOURNAL REFERENCES** | |
| Conference Proceedings: | Workshops, consortia |
| Reference Books/Database: | Encyclopaedia, dictionary, handbook, manuals, databases |
| Industry/Company related documents: | Catalogues, brochures, advertisement, product information, |
| Books: | All books except those categorized as Reference Books |
| Patent related documents: | Legal document, search report, etc. |
| Research / Technical reports: | Technical or research reports of (public) research centres; PhD and master's theses |
| Newspapers/magazines: | Non-scientific, popular |
| Unclear/Others: | Source not identified |

As mentioned above, within the NPL documents sets, scientific articles are usually those most often cited for patents. For this reason, these were taken as the basis for the taxonomy, which is explained below:

Another interesting classification is described by Karvonen and Kässi [10] in Table 2, in which they make an adaptation of the previous taxonomy, but also add a distinction between the references of Science "at large" and Technology "at large". These authors give greater relevance to the references of scientific articles covered by SCI, considering them to be the most scientific, apart from the rest of the literature (other journal literature, conference proceedings or books) which are considered by these authors to be "science at large".

**Table 2.** Taxonomy of non-patent literature (NPL) references [10] (adapted from Callaert et al. [8]).

| SCIENCE "AT LARGE" | |
|---|---|
| SCI-covered journal: | References to scientific publications published in serial journal literature covered by the Science Citation Index (SCI) |
| Not SCI-covered journals: | References to scientific publications published in serial journal literature but NOT covered by the SCI. |
| Conference Proceedings: | Proceedings from conferences and workshops |
| Books (reference books, databases): | All books (including encyclopaedias, handbooks). |
| **TECHNICAL "AT LARGE"** | |
| Industry/company related documents: | Technical disclosure journals and bulletins; company journals: catalogues, brochures; technical reports. |
| Patent related documents: | Patent abstracts; abstract services, search reports |

It is worth noting that in none of the taxonomies are preprints mentioned as a possible citable document, despite being frequently referenced in the academic–scientific field. It is significant that in spite of their exponential growth in scientific repositories, their citation is almost non-existent in patents because they require a minimum of months for processing, and in the meantime, journals have sufficient time for the submission of expert review of these preprints that end up being published articles; hence, the justification for their scarce citation in patents as a type of document.

## 4. Format and Standardization

The format of the citations is regulated by international standards, commonly adopted as Standard ST.14 "Recommendation for the inclusion of references cited in patent documents" of the World Intellectual Property Organization (WIPO). In reference to the NPL citations, in the last revision of 2016, it establishes its bibliographic format according to the International Standard ISO 690:2010 *"Information and documentation—Guidelines for bibliographic references and citations to information resources"*.

This standard also includes the categorization of the citations, assigning them different letters or signs according to the relevance of the document cited in the examined patent, if the citations are of particular importance for the invention or if they only show technological background in general; these categories can be seen in Table 3.

**Table 3.** Categories of document references from World Intellectual Property Organization (WIPO) standard ST. 14.

| | **(a) Categories Indicating Cited Documents (References) of Particular Relevance:** |
| --- | --- |
| Category "X" | The claimed invention cannot be considered novel or cannot be considered to involve an inventive step when the document is taken alone |
| Category "Y" | The claimed invention cannot be considered to involve an inventive step when the document is combined with one or more other such documents, such combination being obvious to a person skilled in the art. |
| | **(b) Categories indicating cited documents (references) of other relevant prior art:** |
| Category "A" | Document defining the general state of the art which is not considered to be of particular relevance |
| Category "D" | Document cited by the applicant in the application and which document (reference) was referred to in the course of the search procedure. |
| Category "E" | Earlier patent document published on or after the international filing date. |
| Category "L" | Document which may throw doubts on priority claim(s) or which is cited to establish the publication date of another citation or other special reason (the reason for citing the document shall be given) |
| Category "O" | Document referring to an oral disclosure, use, exhibition, or other means, for example, conferences proceedings. |
| Category "P" | Document published prior to the filing date (in the case of the PCT, the international filing date) but on or after the priority date claimed in the application. |
| Category "T" | Later document published after the filing date (in the case of the PCT, the international filing date) or priority date and not in conflict with the application but cited to understand the principle or theory underlying the invention |
| Category "&" | Document being a member of the same patent family or document whose contents have not been verified by the search examiner but are believed to be substantially identical to those of another document which the search examiner has inspected |

Although there is an international regulation on format, it is not always applied strictly enough, because this will depend largely on the degree of demand made by each organization on the presentation of bibliographical references in patents. In this sense, some authors [11] have detected obvious problems with the lack of standardization in terms of the purification of data in NPL citations.

In patent information platforms, commercial databases (Derwent Innovation Index on the Web of Science by Clarivate Analytics) or specialized patent search engines (Google

Patents, Lens.org, among others) are already opting for the use of open metadata standards for the extraction of references, while progressively incorporating the use of different persistent identifiers, which helps the bibliographic control of these NPL citations. Furthermore, this automatic citation extraction allows for more structured formats in the records, which facilitates data analysis, as is already the case in scientific–academic information databases. As an example, we have the case of the specialized open search engine Lens.org that gives access to NPL references of patent documents through CrossRef DOI (doi), PubMed ID (pmid), PubMed Central ID (pmcid), Microsoft Academic ID (magid), Core ID (coreid). It also includes ORCID profiles to identify researchers and inventors.

In the case of the databases of the patent offices, despite the fact that they are betting on the optimization of their own datasets to facilitate the dissemination and exchange of their innovations, in the bibliographic field they are still far from the developments carried out by the major search engines such as Google or Bing.

## 5. Cooperation Projects: Common Citation Portals and Open Search Engines

Patent cooperation has been developed over many years by the world's leading patent offices, providing open access to both their own datasets and the shared collections of other offices, such as the Patentscope database of the World Intellectual Property Organization (https://patentscope.wipo.int/, accessed on 30 December 2020) or the ESPACENET database of the European Patent Office (https://worldwide.espacenet.com/, accessed on 30 December 2020).

In the area of citations, there is the FIVE IP OFFICE cooperation project, made up of the offices in Europe, the United States of America, Japan, Korea and China, whose aim is to improve patent examination processes, with services and applications such as the Common Citation Document (CCD) citation consultation tool or the Global Dossier initiative, which provides access to documents associated with the process of granting patents, including lists of bibliographic references submitted by the applicant and examiner, as well as the terms and classifications used in the Prior Art search.

The Global Dossier can be accessed through the USPTO or Espacenet (European Patent Office) portals.

Other open sources of information in which NPL citations can be consulted are the specialized patent search engines that compile large collections of data, such as Google Patents or Lens.org, which, in addition to providing information on NPL references, also have tools for analysing the data, such as maps connecting articles and patents through citation networks in which the influence of scientific research on inventions can be observed.

## 6. Impact of NPL Citations

As mentioned above, through the analysis of NPL citations, indicators can be obtained that measure the impact of these citations from two areas: technological and academic–scientific as can be seen in Figure 1.

In the technological field, the measurement of these references in the patents indicates the degree of scientific intensity in these documents, as well as other aspects such as the inference of scientific research in certain technological sectors, the concentration of NPL citations, the influence of academic institutions, or the level of industrial application of their own research within each country. Therefore:

- NPL citations determine the scientific impact of patents –

In this line, the NPC (non-patent citation) methodology has been established, based on the pioneering work of Carpenter, Narin or Meyer [12], in which various indicators are defined that evaluate the science–technology relationship, such as the Scientific Intensity of Technology indicator that measures the intensity of the use of scientific knowledge in each technology sector "by comparing the average number of citations per patent in a technology sector with the average number of citations per patent in all sectors in the same scientific field". Other interesting indicators provided by these authors are those of the

Technological Diversity of Science, used to find out the concentration of citations of one or several scientific fields in one or several technological sectors.

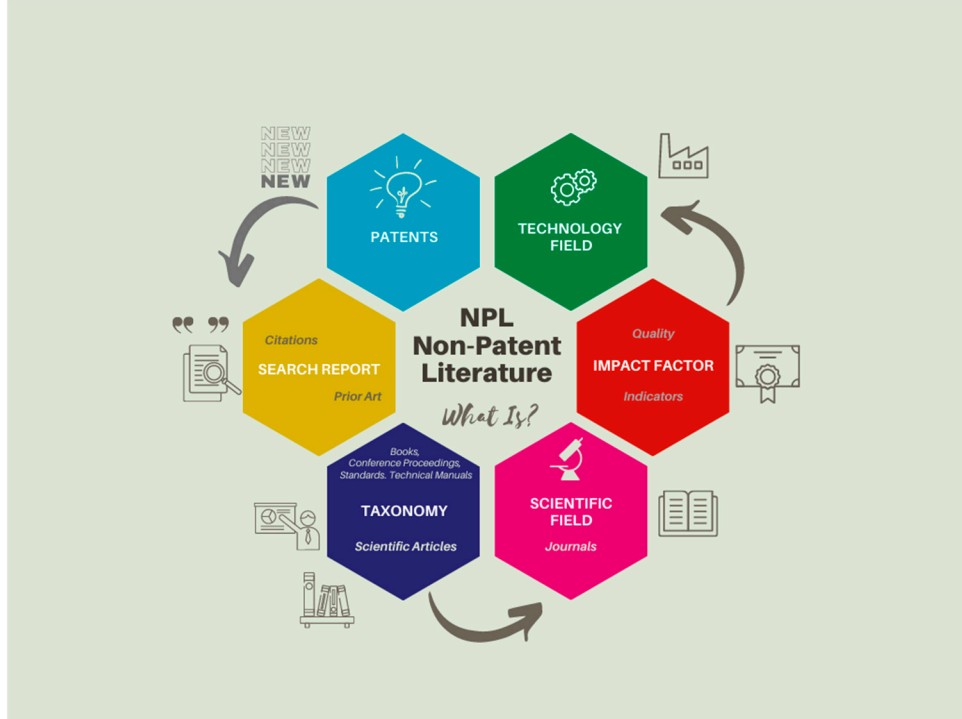

**Figure 1.** Non-patent literature.

From a more individual point of view, the presence of NPL citations on a patent can be a valuable element in relation to its degree of novelty or the impact it may have on other patents. In this sense, there are theories that indicate that, with respect to the degree of novelty, the number of NPL citations determines the topicality of the bibliography used and, therefore, the technology to be patented is considered more novel and cutting-edge. On the other hand, when patents only cite other previous patents, they do not provide such novelty but rather improvements to technology already patented.

If we refer to the degree of impact on other future patents, patents with a greater number of NPL citations, which provide more technological novelty, will have a greater impact on future patents and will be cited more by them, becoming base patents for subsequent technologies.

From the academic–scientific point of view, NPL citations provide information on the impact of scientific production cited in patents, on questions related to the productivity of authors, evaluation of journals, influence and cooperation of research institutions, or citation averages by discipline. Therefore: NPL citations determine the technological impact of scientific publications.

In this sense, one study [13] proposed a technological impact factor (TIF) to evaluate scientific journals in patents, based on the one already established by Journal Citation Report (JCR) in which, according to the author, was assessed by "calculating the number of patents cited to a journal divided by the number of articles published in that particular journal". Other studies [14] focus on the evaluation of the technological impact on specific areas of knowledge, such as the social sciences and humanities, or on the evaluation of the average number of citations per technological sector and per country, to determine the degree of application of each country's own research in the inventions they patent, such as the work of Gazni [15].

There are other studies that investigate NPL citations on very specific aspects, such as establishing a methodology for matching incomplete NPL references in the Scopus and Patstat platforms in order to value scientific productivity in patents [11].

There are also other works that examine NPL applied to a specific subject, as is the case of the study by Velayos-Ortega and Lopez-Carreño [16], in which the authors analyse the citations of scientific publications in the patents on the novel coronavirus disease 2019 in the specialized search engine Lens.org, obtaining, among other results, a ranking of the most cited journals in these patents, on which they make a comparison with their positioning in JCR of the Web of Science.

In the establishment of indicators of the technological impact of the scientific publications referenced in the patents, it must be taken into account that the patents are not published as quickly as the research, because usually, a minimum of 18 months must pass from the moment from which a patent application is submitted until it is made public. This is similar to the case with citations of these documents, the dynamics of which are slower than those of scientific documents, as indicated by the study [17], a slowdown to be taken into account for impact metrics of scientific publications cited by patents, especially those intended to measure immediacy.

From this perspective, the measurement of NPL citations with indicators different from traditional bibliometrics could be considered as an assessment element to be taken into account in science and technology evaluation systems, but previously it requires treatment, standardization and bibliographic management from the issuing offices, in the same way as occurs with scientific publications, and more specifically in journals.

**Author Contributions:** Conceptualization, G.V.-O. and R.L.-C.; methodology, G.V.-O.; investigation, G.V.-O.; resources, R.L.-C.; writing—original draft preparation, G.V.-O.; writing—review and editing, G.V.-O. and R.L.-C.; supervision, R.L.-C. Both authors have read and agreed to the published version of the manuscript.

**Funding:** This research received no external funding.

**Conflicts of Interest:** The authors declare no conflict of interest.

**Entry Link on the Encyclopedia Platform:** https://encyclopedia.pub/4307.

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
