# Peer review of "Non-Patent Literature"

_encyclopedia, doi:10.3390/encyclopedia1010019_

Round 1

Reviewer 1 Report

The article is well organized and very useful for scientific communities. I think the article will be accepted with minor corrections. Please consider the following comments.

1)How is the relationship with the following article?

https://encyclopedia.pub/4307

2)Tables 1 and 2:

The authors mention on SCI covered journal. How about SSCI (Social Science Citation) Journal?

3)Tables 1 and 2:Industry / Company related documents

How about public documents published by central /local governments?

Author Response

English language and style:

  • Reviewed spell check

Comments and Suggestions for Authors:

The article is well organized and very useful for scientific communities. I think the article will be accepted with minor corrections. Please consider the following comments.

1)How is the relationship with the following article?

https://encyclopedia.pub/4307

  • This document is an extension of the entry: https://encyclopedia.pub/4307

2) Tables 1 and 2:

The authors mention on SCI covered journal. How about SSCI (Social Science Citation) Journal?

  • This classification refers to the one established in the work of Julie Callaert; Bart Van Looy; Arnold Verbeek; Koenraad DeBackere; Bart Thijs; Traces of Prior Art: An analysis of non-patent references found in patent documents. Scientometrics 2006, 69, 3-20, 10.1007 / s11192-006-0135-8 where SSCI is not mentioned.

3) Tables 1 and 2: Industry / Company related documents

How about public documents published by central /local governments?

  • Industry / Company related documents refer to technical working and procedural documents that are sometimes documentary sources cited in patents. This taxonomy is the one established by authors Callaert et al. in which documents published by central /local governments are not mentioned. A more exhaustive description of all the categories of documents cited in patents would be the subject of another study.

Reviewer 2 Report

The paper is a review of Non-Patent Literature. It covers all denotations and definitions, describes different types of NPL, gives examples of use etc. In this regard, the paper is absolutely appropriate. It should only

- describe in more detail how to use ESPACENET for NPL searches,

- cite the pioneer work Mark P. Carpenter, Martin Cooper & Francis Narin (1980) Linkage Between Basic Research Literature and Patents, Research Management, 23:2, 30-35.

Author Response

English language and style:

  • Reviewed spell check

Comments and Suggestions for Authors:

The paper is a review of Non-Patent Literature. It covers all denotations and definitions, describes different types of NPL, gives examples of use etc. In this regard, the paper is absolutely appropriate. It should only

- describe in more detail how to use ESPACENET for NPL searches,

  • We mention Espacenet database for its contents (data collections) and services (access to Common Citation Document and Global Dossier). The search for NPL references in this database would be the subject of another study.

- cite the pioneer work Mark P. Carpenter, Martin Cooper & Francis Narin (1980) Linkage Between Basic Research Literature and Patents, Research Management, 23:2, 30-35.

  • We've done it. We have added the suggested citation.

Round 2

Reviewer 1 Report

The manuscript has been well improved.